# Enhanced 15-Lipoxygenase 1 Production is Related to Periostin Expression and Eosinophil Recruitment in Eosinophilic Chronic Rhinosinusitis

**DOI:** 10.3390/biom10111568

**Published:** 2020-11-18

**Authors:** Yoshimasa Imoto, Tetsuji Takabayashi, Masafumi Sakashita, Yukinori Kato, Kanako Yoshida, Masanori Kidoguchi, Keisuke Koyama, Naoto Adachi, Yukihiro Kimura, Kazuhiro Ogi, Yumi Ito, Masafumi Kanno, Masayuki Okamoto, Norihiko Narita, Shigeharu Fujieda

**Affiliations:** Department of Otorhinolaryngology Head & Neck Surgery, Faculty of Medical Sciences, University of Fukui, Fukui 910-1193, Japan; tetsuji@u-fukui.ac.jp (T.T.); msaka@u-fukui.ac.jp (M.S.); ykato@u-fukui.ac.jp (Y.K.); kosikana@u-fukui.ac.jp (K.Y.); kidoguti@u-fukui.ac.jp (M.K.); fukui.ent.kei@gmail.com (K.K.); adachin@u-fukui.ac.jp (N.A.); kimyuki@u-fukui.ac.jp (Y.K.); ogikazu@u-fukui.ac.jp (K.O.); rbyumi@yahoo.co.jp (Y.I.); pradaclasse@yahoo.co.jp (M.K.); masayu@u-fukui.ac.jp (M.O.); norihiko@u-fukui.ac.jp (N.N.); sfujieda@u-fukui.ac.jp (S.F.)

**Keywords:** CRS, 15-LOX-1, eosinophils, periostin, IL-33

## Abstract

Background: The pathological features of chronic rhinosinusitis (CRS) with nasal polyps (CRSwNP) tissues include an eosinophilic infiltration pattern (eosinophilic CRS (ECRS)) or a less eosinophilic pattern (non-ECRS). Recently, it has been suggested that 15-lipoxygenase 1 (15-LOX-1) may have significant roles in allergic disease; however, the significance of 15-LOX-1 in CRS is not well understood. The objective of this study was to demonstrate the expression of 15-LOX-1 in CRS. Methods: The mRNA expression levels of 15-LOX-1 and periostin in nasal tissues were measured by quantitative real-time polymerase chain reaction. We also performed an immunofluorescence study of nasal tissues. Cells of the Eol-1 eosinophilic leukemic cell line were stimulated with interleukin-33 to test the induction of 15-LOX-1. Results: The expression level of 15-LOX-1 mRNA in nasal polyps (NPs) was significantly higher in ECRS patients than in non-ECRS patients. The immunofluorescence study revealed that both airway epithelial cells and eosinophils in NPs expressed 15-LOX-1. A significant correlation was seen between the number of eosinophils and the mRNA expression levels of 15-LOX-1 and periostin in nasal polyps. Moreover, interleukin-33 enhanced 15-LOX-1 expression in Eol-1 cells. Conclusions: 15-LOX-1 was shown to be a significant molecule that facilitates eosinophilic inflammation in ECRS.

## 1. Introduction

Chronic rhinosinusitis (CRS) is a disease that is defined by inflammation of the nose and paranasal sinuses that lasts for at least 12 weeks and causes nasal obstruction, facial pain, olfactory dysfunction, and mucopurulent drainage [1,2,3]. CRS is one of the most common chronic diseases with high comorbidity of asthma [4,5,6]. Clinically, CRS is frequently divided into two groups based on the presence or absence of nasal polyps (NPs), namely CRS with NPs (CRSwNP) and CRS without NPs (CRSsNP). Each type of CRS describes a heterogeneous spectrum [7,8,9]. Pathologically, in Western countries, CRSwNP tissues often display intense eosinophilic infiltration and a type2 biased cytokine profile [8,10]; in contrast, in Asia, they are more often characterized by neutrophil-dominant inflammation [11,12]. However, the proportion of type 2-signature CRS has been increasing in several Asian countries [13,14]. A multicenter retrospective study in Japan named the Japanese Epidemiological Survey of Refractory Eosinophilic Chronic Rhinosinusitis (JESREC) examined the clinical and pathological features of 1716 patients who underwent sinus surgery [15]. According to the JESREC study, a definite diagnosis of eosinophilic CRS (ECRS) can be made when the mean number of eosinophils is 70 or more in at least three high-power fields. The study showed that blood eosinophils, ethmoid disease, asthma, and aspirin intolerance were all associated with disease recurrence and the need for further surgical intervention. In general, topical and oral corticosteroids are the mainstays of therapy for patients with ECRS. Sinus surgery therapy is also one of the effective therapies. Although patients with non-ECRS have a low recurrence rate after surgery, a significant proportion of patients with ECRS require repeated sinus surgery due to recurrence despite these medical and/or surgical therapies. Thus, elucidation of the precise molecular mechanisms is required to better understand the pathological differences between ECRS and non-ECRS.

12/15-Lipoxygenase (15-LOX-1) has a significant role in inflammatory processes [16]. 15-LOX-1 catalyzes the oxygenation of polyunsaturated fatty acids by inserting molecular oxygen at the C15 position of arachidonic acid or the C13 position of linoleic acid to produce 15S-hydroxyeicosatetraenoic acid or eoxins [17,18], which are potent pro-inflammatory metabolites that have a significant role in the pathogenesis of asthma and aspirin-exacerbated respiratory disease (AERD) [19]. Li et al. reported that the 15-LOX-1 mRNA expression level is elevated in NP tissues when compared to the inferior turbinate or middle turbinate in the nasal cavity [20]. A genome-wide association study of patients with NPs across several cohorts showed that a single nucleotide polymorphism in the *15-LOX-1* gene was associated with a reduction in the risk of NPs, suggesting the significance of 15-LOX-1 in the pathogenesis of NPs [21]. Furthermore, a single-cell RNA sequencing analysis revealed dysregulated arachidonic metabolism in the 15-LOX-1 pathway in patients with AERD [22]. It has been considered that 15-LOX-1 may have important roles in the pathogenesis of NPs; however, the precise mechanisms are not yet well understood. In the present study, we compared the expression levels of 15-LOX-1 in the NPs in ECRS and non-ECRS subjects. We also examined the expression of 15-LOX-1 in eosinophils in the NPs.

## 2. Materials and Methods

### 2.1. Patient Recruitment and Clinical Sample Collection

Patients with ECRS or non-ECRS were recruited from the Department of Otorhinolaryngology Head and Neck Surgery, University of Fukui, Japan. All subjects provided written informed consent. The protocols and procedures for the study were approved by the Ethics Committee of the University of Fukui, and they are in compliance with the Declaration of Helsinki and Good Clinical Practice. Uncinate tissues (UTs) and NP tissues were obtained from routine functional endoscopic sinus surgery in patients with ECRS or non-ECRS. Control specimens from patients without CRS were obtained from patients with sinonasal cyst, inverted papilloma, or nasal septum deviation. Patients with an established immunodeficiency, pregnancy, coagulation disorder, diagnosis of classic allergic fungal sinusitis, Churg–Strauss syndrome (eosinophilic granulomatosis with polyangiitis), or cystic fibrosis were excluded from the study. All patients scheduled for surgery had previously failed to respond to adequate trials of conservative medical therapy (prolonged antibiotic regimens, nasal steroid sprays, oral steroids, saline irrigations, and decongestants) for the control of symptoms. All subjects were prohibited from taking oral steroids for at least 4 weeks prior to surgery. AERD was diagnosed based upon clinical criteria and was defined by the presence of asthma and at least one hypersensitivity reaction, including nasal congestion, or shortness of breath within 2–3 h of ingestion of either aspirin or NSAIDs. To count the proportions of white blood cells, whole blood cells were collected with EDTA before surgery. The detailed patient characteristics are shown in Table 1. All data are presented as the mean ± standard error of the mean (SEM).

### 2.2. Histological Analysis

The NP tissues from patients with ECRS or non-ECRS were obtained during surgery. Tissues were immediately fixed in 10% formalin, embedded in paraffin, and cut into thin sections. The sections were stained with hematoxylin-eosin. The number of eosinophils in the mucosa was counted in three high-power fields (×400) within the three densest areas with cellular infiltration beneath the epithelial surface. The mean number of eosinophils was calculated. Histological examinations were performed by three expert doctors who were blinded to the clinical data as previously reported [15].

### 2.3. Real-Time Polymerase Chain Reaction (PCR)

The nasal tissues were immediately placed in a stabilization reagent (RNA later, Thermo Fisher Scientific, Waltham, MA, USA), and total RNA was extracted using NucleoSpin RNA II (Macherey-Nagel, Bethlehem, PA, USA) with DNase I (Invitrogen, Carlsbad, CA, USA) according to the manufacturers’ instructions. The quality of the total RNA from sinus tissues was assessed with a 2100 Bioanalyzer (Agilent Technologies, Santa Clara, CA, USA) using a RNA 6000 Nano LabChip (Agilent Technologies, Santa Clara, CA, USA). Single-strand cDNA was synthesized with a High Capacity cDNA Reverse Transcription Kit (Thermo Fisher Scientific, Waltham, MA, USA). Semi-quantitative real-time reverse transcription PCR (RT-PCR) was performed with the TaqMan method using an Applied Biosystems StepOnePlus Real-Time PCR system (Thermo Fisher Scientific, Waltham, MA, USA) in 15 μL reaction volumes (7.5 μL of 2× TaqMan Master mix (Thermo Fisher Scientific, Waltham, MA, USA) and 0.75 μL of 20× primer and probe mixture). Probes for 15-LOX-1, periostin, and glyceraldehyde 3-phosphate dehydrogenase (GAPDH) were purchased from Thermo Fisher Scientific. Aliquots of cDNA equivalent to 10 ng of total RNA were used for real-time PCR. The mRNA expression levels were normalized to the median expression of the *GAPDH* housekeeping gene.

### 2.4. Immunohistochemistry

The NP tissues were immediately fixed in 10% formalin, embedded in paraffin, and sectioned into 3 μm slices using a Retoratome REM-710 (Yamato Kohki, Saitama, Japan). Blocked sections were incubated with mouse anti-human 15-LOX-1 monoclonal antibody (mAb; LS-B12236, LifeSpan Bioscience, Seattle, WA, USA) at a dilution of 1:1000 overnight at 4 °C. After washing, sections were incubated with ABC reagent (Vector Laboratories, Burlingame, CA, USA) for 1 h. Subsequently, the sections were rinsed and incubated in DAB reagent (Thermo Fisher Scientific, Waltham, MA, USA), then counterstained with hematoxylin. The sections were then dehydrated, cleaned, and mounted on slides with coverslips using PARAmount-N (FALMA, Osaka, Japan). Microscopic analysis was performed with an Olympus BX53 upright research microscope using 20× and 40× objective lenses (Olympus, Tokyo, Japan), and images were collected with cellSens (Olympus, Tokyo, Japan). For the semiquantitative analysis of 15-LOX-1, slides were blinded, 10 photographic fields were randomly taken from each slide and then rated on a scale of 0 to 3 by a blinded observer. A rating of 0 indicated no staining, 1 indicated mild staining, 2 indicated moderate staining, and 3 indicated intense staining.

For the immunofluorescence assay, rehydrated sections were blocked with blocking buffer (Agilent Technologies, Santa Clara, CA, USA). For 15-LOX-1 and eosinophil cationic protein (ECP) staining, tissue sections were incubated with 15-LOX-1 mAb (LS-B12236, LifeSpan Bioscience, Seattle, WA, USA; 1:1000) and rabbit anti-human ECP polyclonal antibody (bs-8615R, Bioss Antibodies, Woburn, MA, USA; 1:100) in a blocking buffer overnight at 4 °C. After washing with phosphate-buffered saline, cells were incubated with 4 μg/mL Alexa Fluor 488-conjugated donkey anti-mouse IgG (Thermo Fisher Scientific, Waltham, MA, USA) and 4 μg/mL Alexa Fluor 568-conjugated donkey anti-rabbit IgG (Thermo Fisher Scientific, Waltham, MA, USA) for 1 h at room temperature in the dark. After a final washing with phosphate-buffered saline, coverslips were placed onto the slides with SlowFade Gold anti-fade reagent with 49,6-diamidino-2-phenylindole (Thermo Fisher Scientific, Waltham, MA, USA), and the slides were stored in the dark at 4 °C. Images from the immunofluorescence slides were obtained with an Olympus BX53 inverted research microscope (Olympus, Tokyo, Japan), and images were collected with cellSens software (Olympus, Tokyo, Japan).

### 2.5. Cell Culture and Treatments

Primary normal human bronchial epithelial (NHBE) cells were purchased from Lonza (Walkersville, MD, USA). We used NHBE cells from at least three different donors. NHBE cells were seeded in collagen-coated 24-well plates and were maintained in serum-free bronchial epithelial cell growth medium (BGEM, Lonza, Walkersville, MD, USA). Before stimulation, the NHBE cells were cultured in BEGM without hydrocortisone for 24 h. At confluency, submerged NHBE cells were stimulated with poly(I:C) (R&D Systems, Minneapolis, MN, USA), interferon (IFN)-γ (R&D Systems, Minneapolis, MN, USA), and interleukin (IL)-4 (R&D Systems, Minneapolis, MN, USA) for 24 h. Then, total RNA was isolated from the cells.

The Eol-1 human eosinophilic leukemic cell line was obtained from the Riken Cell Bank (Tsukuba, Japan). The cells were maintained in RPMI1640 medium (Gibco Laboratories, Grand Island, NY, USA) supplemented with 10% defined fetal bovine serum (Gibco) and 20 mM Hepes buffer (Gibco) in a humidified atmosphere with 5% CO_2_ at 37 °C as described previously [23]. For the cytokine stimulation study, 5 × 10^5^ Eol-1 cells were incubated with a varying concentration of IL-5 (R&D Systems, Minneapolis, MN, USA) and IL-33 (R&D Systems, Minneapolis, MN, USA) for 24 h. Total RNA from the cultured Eol-1 cells was extracted, and single-strand cDNA was synthesized.

### 2.6. Statistical Analysis

All data are presented as the mean ± standard error of the mean (SEM) unless otherwise mentioned. The data from the culture experiments were analyzed using a paired Student’s *t* test. Differences between groups were analyzed by a Kruskal-Wallis analysis of variance (ANOVA) with Dunnett’s post hoc testing, Wilcoxon signed-rank test, and a Mann–Whitney U test. In all cases, *p* < 0.05 was considered to be statistically significant. All statistical analyses were performed using GraphPad Prism 5.0 (GraphPad Software, La Jolla, CA, USA) software.

## 3. Results

### 3.1. 15-LOX-1 mRNA Expression was Upregulated in NP Tissues from the ECRS Patients but not Those from the Non-ECRS Patients

Ninety-three patients were included in the study. We included UT samples from control subjects without CRS as well as NP samples from CRSwNP subjects, which included both ECRS and non-ECRS subjects. The comorbidity of asthma was significantly more common in ECRS (NPs) subjects than in CRS (UTs) and non-ECRS (NPs) subjects. As for the proportion of eosinophils in peripheral blood, ECRS (NPs) subjects showed more elevated eosinophils levels than the other groups. Table 1 shows the characteristics of the subjects in the study. The mRNA from whole tissue extracts was analyzed to compare the expression levels of 15-LOX-1. As shown in Figure 1, the level of 15-LOX-1 mRNA was significantly higher in the NPs of the ECRS patients than in the control UTs, CRS UTs, and NPs from the non-ECRS patient (vs. control UT, *p* < 0.0001; vs. CRS UT, *p* < 0.0001; vs. NPs from non-ECRS subjects, *p* < 0.0001). In contrast, the expression level of 15-LOX-1 in NPs from the non-ECRS subjects was not elevated when compared to those of the control UTs and CRS UTs.

### 3.2. Both Epithelial Cells and Eosinophils Expressed 15-LOX-1 in Sinonasal Tissues

To assess the expression of 15-LOX-1 in sinonasal tissues and in NP tissues, we performed an immunohistochemistry analysis of the surgical specimens of UTs from the controls and NPs from both the ECRS and non-ECRS subjects (Figure 2A–D). We found that 15-LOX-1 was expressed in epithelial cells, which is in line with a previous report [20]. Cellular staining was graded by blinded observers for intensity. This semiquantitative analysis showed significantly more intense 15-LOX-1 staining in NPs from patients with ECRS compared with staining seen in UT from control subjects (*p* < 0.001), in UT from patients with CRS (*p* < 0.0001), or in NPs from patients with non-ECRS (*p* < 0.0001) (Figure 2E). Interestingly, we found that 15-LOX-1 was also expressed in inflammatory cells that infiltrated into the NPs in ECRS subjects. Since we expected that 15-LOX-1-expressing inflammatory cells might be eosinophils, we then performed an immunofluorescence assay for 15-LOX-1 and ECP, which is a marker for eosinophils [9]. The results showed that 15-LOX-1 was also expressed in eosinophils (Figure 3).

### 3.3. 15-LOX-1 mRNA Expression was Correlated with Eosinophilic Inflammation in Nasal Tissues

As the immunofluorescence assay revealed that eosinophils expressed 15-LOX-1, we then investigated the correlation between the number of eosinophils and the mRNA expression level of 15-LOX-1 in NPs. Interestingly, we found a significant correlation between the number of eosinophils and the mRNA expression level of 15-LOX-1 in NPs (all CRS subjects, r = 0.5987, *p* < 0.0001, Figure 4A; only ECRS subjects, r = 0.4102, *p* < 0.01, Figure 4B). We also observed a significant correlation between the proportion of peripheral blood eosinophils and mRNA expression level of 15-LOX-1 in NPs (r = 0.4428, *p* < 0.001, data not shown).

Periostin is a matricellular protein that plays significant roles in allergic airway diseases, such as asthma [24,25,26] and CRS [27,28]. Periostin has been considered a promising biomarker in allergic disease. We looked at the correlation between the mRNA expression levels of 15-LOX-1 and periostin in sinonasal tissues. As shown in Figure 4C, there was a significant correlation between the mRNA expression levels of 15-LOX-1 and periostin (r = 0.7540, *p* < 0.0001), especially only in ECRS subjects (r = 0.8297, *p* < 0.0001, Figure 4D). We also found a significant correlation between the expression levels of periostin and the number of eosinophils in NPs (r = 0.588, *p* < 0.0001, *n* = 71). These results indicate that an elevated 15-LOX-1 level is closely related to eosinophilic inflammation in nasal tissues.

### 3.4. IL-4, but not IFN-γ nor poly(I:C), Induced 15-LOX-1 from NHBE Cells

Since the immunohistochemistry analysis showed that epithelial cells expressed 15-LOX-1 in sinonasal tissues, especially in the NPs from ECRS patients, we investigated the induction of 15-LOX-1 in airway epithelial cells. We stimulated submerged cultured NHBE cells with IL-4, IFN-γ, and poly(I:C) in vitro. As shown in Figure 5, IL-4 strongly induced 15-LOX-1 expression (*p* < 0.01), whereas IFN-γ and poly(I:C) did not.

### 3.5. IL-33 Induced 15-LOX-1 from Eol-1 Cells

We demonstrated the induction of 15-LOX-1 by using Eol-1 cells. Analysis of mRNA expression by real-time RT-PCR revealed that EoL-1 cells constitutively expressed 15-LOX-1 (Figure 6). IL-33, a member of the IL-1 family of cytokines, is expressed by various cells, including nasal tissues such as epithelial cells, endothelial cells, fibroblasts, and smooth muscle cells [29,30,31]. IL-33 has emerged as a mediator that orchestrates the pathogenesis of airway eosinophil-related allergic disease, including asthma and CRS [32,33,34]. Reportedly, IL-33 plays a pivotal role in the chronic inflammation of CRSwNP by enhancing Th2 cytokine production, which further promotes the Th2 profile in CRSwNP [33,34,35,36]. Interestingly, enhanced expression of IL-33 mRNA and protein has been found in the sinus tissues of patients with CRSwNP [34]. Next, we stimulated cultured Eol-1 cells with IL-33 or IL-5 in vitro. As shown in Figure 6A, IL-33 enhanced 15-LOX-1 expression in Eol-1 cells in a concentration-dependent manner, but IL-5 did not (Figure 6B). Taken together, these results indicated that the elevated 15-LOX-1 level in NP tissues from ECRS subjects was due to the stimulation of both epithelial cells and eosinophils in NPs by Th2-biased pro-inflammatory cytokines, such as IL-4 and IL-33.

## 4. Discussion

Many types of inflammatory cells infiltrate into NP tissues, which display several patterns of inflammatory responses. For the endotyping of CRS, the measurement of cytokines and chemokines is useful [9,37,38]. Emerging evidence has indicated that several biologics targeting Th2 cytokines as well as IgE show promise as therapeutics for severe and uncontrolled ECRS [39]. Thus, for appropriate, effective, and successful therapy, endotyping of CRS is important [9,38,39,40]. Recent reports using gene expression analysis and a genome-wide association study have revealed the significance of 15-LOX-1 in the pathogenesis of CRSwNP [20,21,41]. Although the appearance of NPs may be similar, the pathophysiological features of NPs show heterogeneity and are complicated [7,39]. The JESREC study could distinguish ECRS (eosinophilic inflammation) and non-ECRS (neutrophilic inflammation) based on both the pathological and clinical features. A remarkable finding of the present study is that the gene expression level of 15-LOX-1 in NPs can clearly distinguish ECRS and non-ECRS (Figure 1). This may be because neutrophil expresses 15-LOX-2 but not 15-LOX-1 [42]. Since one of the most important clinical features of ECRS, as defined by the JESREC study, is a high recurrence ratio of NPs when compared to non-ECRS after surgery, the present finding may also have significant value for choosing a therapeutic strategy.

Periostin is highly deposited in the subepithelial regions of many chronic inflammatory diseases. Periostin activates both immune and non-immune pathways, which further augments inflammation [24,25]. Several reports have revealed that IL-4 and IL-13, signature type 2 cytokines, can induce periostin expression. In eosinophilic chronic disease, periostin facilitates the adhesion and migration of eosinophils [24,26]. As for ECRS, Xu et al. reported that periostin is related to the onset and prognosis of ECRS [43]. Ninomiya et al. showed that the serum levels of periostin were upregulated in ECRS subjects, especially in moderate and severe ECRS subjects [27]. They also revealed that NPs in patients with ECRS showed elevated levels of periostin deposition. Their results also demonstrated that the polyp recurrence rate was significantly higher in patients with a high serum periostin level, which indicated that the preoperative serum periostin level could be a clinical marker for recurrence after surgery. Our present results revealed that the mRNA expression level of 15-LOX-1 was significantly correlated to that of periostin in NPs. Furthermore, 15-LOX-1 mRNA expression had a significant correlation with the number of eosinophils in NPs. These results suggest that an elevated 15-LOX-1 level in the NPs of ECRS subjects is closely related to eosinophilic inflammation.

The IL-33 receptor, a heterodimeric complex composed of T2 and IL-1 receptor accessory protein (IL1RAP), is expressed on Th2 cells, mast cells, basophils, eosinophils, and macrophages [30,31,44]. IL-33 can activate many types of immune cells via its receptor, interleukin receptor-like 1 (IL1RL1, also known as ST2). Reportedly, an increased expression level of ST2 was observed in the sinus mucosa of CRSwNP subjects [33]. In airway allergic disease, IL-33 stimulation of eosinophils enhances membrane ST2 expression [45]. Although IL-33 is considered to play significant roles in NPs, the precise mechanism of the effect of IL-33 on eosinophils in NPs remains unclear. Proteases, which are contained in antigens, bacteria, and fungi, can induce IL-33 from epithelial cells and thus promote the Th2 response [46,47]. CRSwNP is associated with a high rate of nasal colonization by *Staphylococcus aureus* (*S. aureus*), which is associated with a severe disease phenotype, and *S. aureus* superantigens have been implicated in the pathogenesis of the Th2 response in the nasal mucosa as they induce the release of several cytokines such as IL-33 [36,48,49,50]. Recent reports have shown that group 2 innate lymphoid cells (ILC2s), which also express ST2, play significant roles in allergic diseases, such as asthma and CRSwNP [33,51,52]. ILC2 stimulated by epithelia-derived IL-33 secrete Th2 cytokines, such as IL-4 and IL-13 [33]. These cytokines activate and enhance the survival of other inflammatory cells, such as eosinophils, basophils, and mast cells, which further promotes cell infiltration and inflammation in NPs. IL-4 then facilitates 15-LOX-1 expression in epithelial cells (Figure 5), which enhances CCL26 expression and eosinophil infiltration into NPs [20]. An important finding of the present study is that IL-33 enhanced 15-LOX-1 expression in Eol-1 cells (Figure 6A), but IL-5 did not (Figure 6B). This finding may indicate that epithelia-derived IL-33 can directly and/or indirectly accelerate eosinophil infiltration into NPs via elevated 15-LOX-1 levels; this notion may be supported by the result that 15-LOX-1 mRNA expression was significantly correlated with the number of eosinophils (Figure 4A,B) and the expression of periostin (Figure 4C,D) in NP tissues. Altogether, our results reveal that IL-33 contributes to the enhancement of 15-LOX-1 expression in eosinophils, which may lead to an eosinophilic inflammatory shift (shift to ECRS). Emerging evidence has shown that several biologics targeting Th2 cytokines, including IL-4, show promise as therapeutics for severe and uncontrolled ECRS [53,54]. Anti-IL-33 biologics may be considered as candidates for the treatment of ECRS in the future.

## 5. Conclusions

In conclusion, the expression level of 15-LOX-1 was elevated in the NPs from ECRS subjects, but not in those from non-ECRS subjects. 15-LOX-1 was expressed in both nasal epithelial cells and eosinophils in the NPs, and its expression was enhanced by the Th2 inflammatory response. Epithelial and subepithelial Th2 inflammatory signatures can further amplify eosinophilic inflammation via elevated 15-LOX-1 levels. As 15-LOX-1 may have a crucial role in the pathogenesis of ECRS, 15-LOX-1 may be a new candidate molecule for the treatment of ECRS.

## Figures and Tables

**Figure 1 biomolecules-10-01568-f001:**
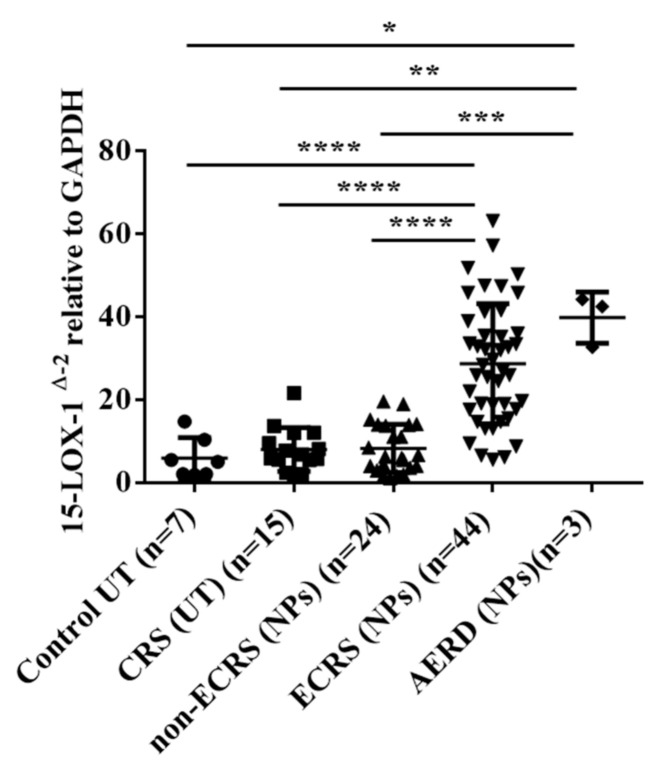
Total RNA was extracted from the nasal UTs and NP tissues. The expression of 15-LOX-1 mRNA was analyzed by RT-PCR. The expression levels of mRNA were normalized to those of the glyceraldehyde 3-phosphate dehydrogenase housekeeping gene (*GAPDH*). * *p* < 0.05, ** *p* < 0.01, *** *p* < 0.001, and **** *p* < 0.0001. (●: control UTs; ■: UTs from CRS subjects; ▲: NP tissues from non-ECRS subjects; ▼: NP tissues from ECRS subjects).

**Figure 2 biomolecules-10-01568-f002:**
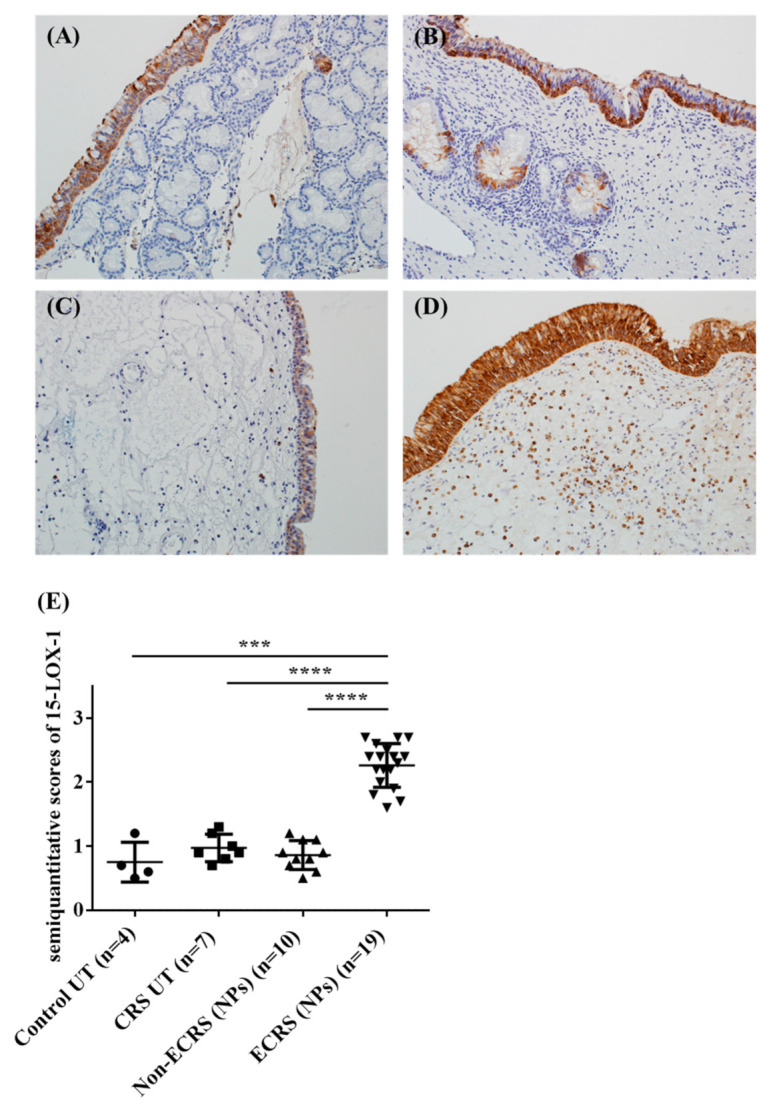
Immunohistochemical (IHC) detection of 15-LOX-1 (**A**–**D**) in human sinonasal tissues. Representative immunostaining for 15-LOX-1 in the UT from a control subject (**A**), in the UT from an ECRS subject (**B**), in the NP from a non-ECRS subject (**C**), and in the NP from an ECRS subject (**D**). Semiquantitative analysis of 15-LOX-1 in UT from control subjects (*n* = 4), in UT from patients with CRS (*n* = 7), in NP from patients with non-ECRS (*n* = 10), and in NP from patients with ECRS (*n* = 19) was performed (**E**). Magnification: ×200. *** *p* < 0.001, and **** *p* < 0.0001.

**Figure 3 biomolecules-10-01568-f003:**
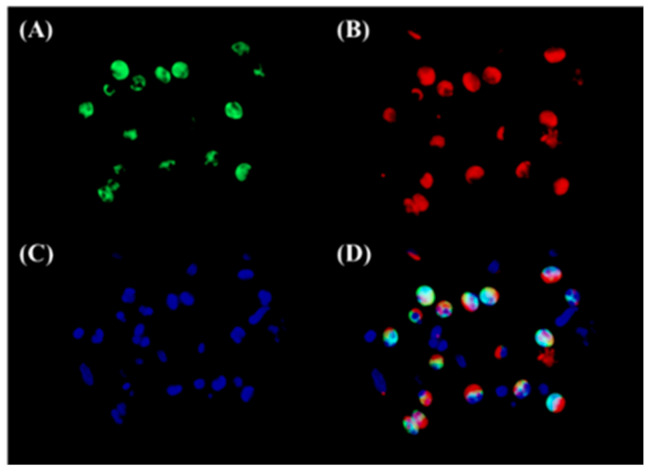
Immunofluorescence of 15-LOX-1 and ECP in the sub-epithelial lesion of NP tissues from ECRS patients. An immunofluorescence assay was performed with anti-15-LOX-1 antibody (green fluorescence) (**A**), and anti-ECP mAb (red fluorescence) for eosinophils (**B**). Nuclei were counterstained with 4′,6-diamidino-2-phenylindole (DAPI); blue fluorescence, (**C**); and the signals were merged (**D**). The results are representative images from six different patients. Magnification: ×400.

**Figure 4 biomolecules-10-01568-f004:**
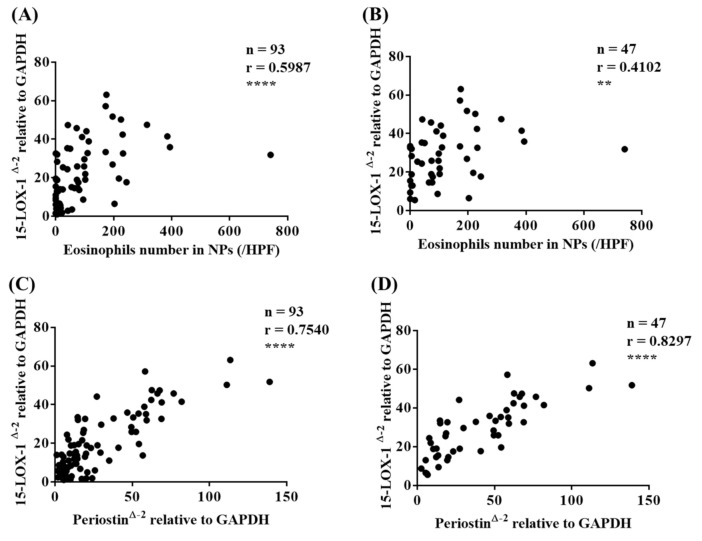
The correlation between 15-LOX-1 mRNA expression and the number of eosinophils in NP tissues (all CRS subjects, (**A**); ECRS subjects, (**B**)). The correlation between 15-LOX-1 and periostin mRNA expression in nasal tissues (all subjects, (**C**); ECRS subjects, (**D**). The correlations were assessed using a Spearman rank correlation test. ** *p* < 0.01, and **** *p* < 0.0001.

**Figure 5 biomolecules-10-01568-f005:**
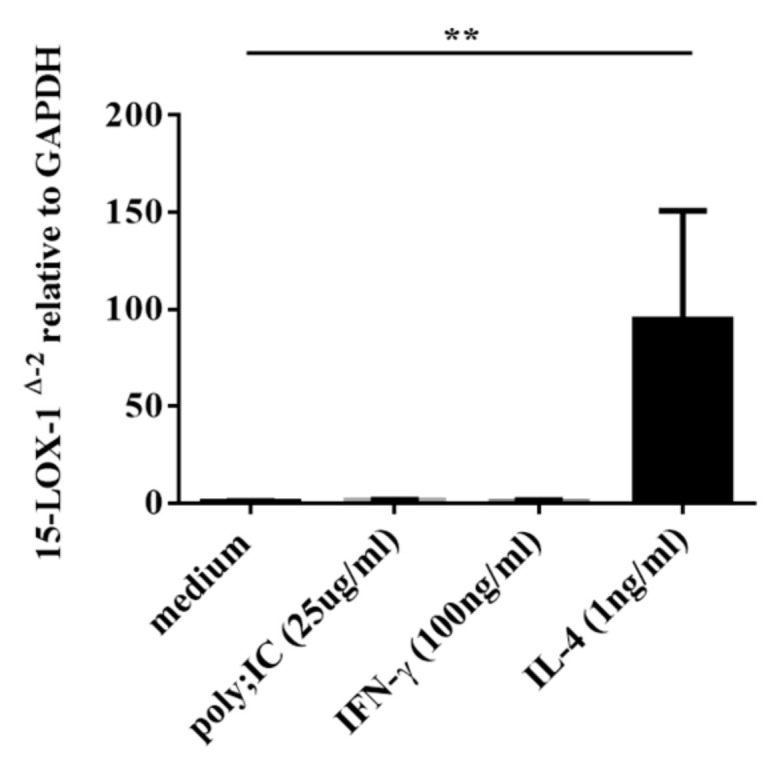
Submerged normal human bronchial epithelial (NHBE) cells were stimulated with poly(I:C) (25 µg/mL), IFN-γ (100 ng/mL), and IL-4 (1 ng/mL) for 24 h. (A) Cell lysates were harvested for RNA extraction to analyze 15-LOX-1 mRNA expression by RT-PCR (*n* = 3–6). Data shown are the means ± SEM of four independent experiments. ** *p* < 0.01 when compared to non-stimulated cells.

**Figure 6 biomolecules-10-01568-f006:**
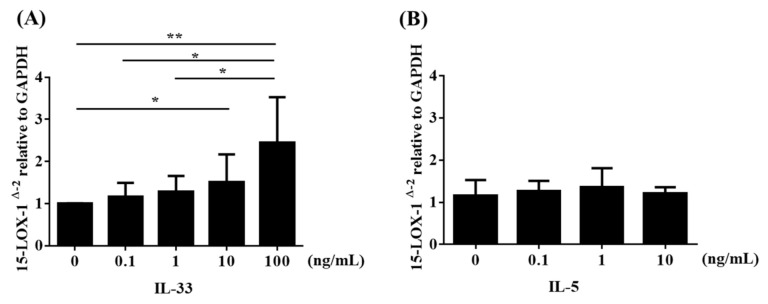
Eol-1 cells were stimulated with (**A**) IL-33 or (**B**) IL-5 for 24 h. Cell lysates were harvested for RNA extraction to analyze 15-LOX-1 mRNA expression by RT-PCR (*n* = 4–6). Data shown are the means ± SEM of independent experiments.* *p* < 0.05 and ** *p* < 0.01 when compared to the non-stimulated cells. NS, not significant.

**Table 1 biomolecules-10-01568-t001:** Characteristics of the subjects.

Subjects	Control UT (*n* = 7)	CRS (UT) (*n* = 15)	Non-ECRS (NPs) (*n* = 24)	ECRS (NPs) (*n* = 7)
Gender, male/female	6/1	9/6	18/6	35/13
Mean ± SEM, Age, years	52.7 ± 19.4	49.3 ± 18.0	50.5 ± 15.6	54.7 ± 13.6
Asthma, yes/no	2/5	2/13	1/23	22/26 ^#,†††^
AERD, yes/no	0/7	0/15	0/24	3/45
Mean ± SEM Serum total IgE (IU/mL)	287.6 ± 324.4	743.2 ± 1390.1	322.7 ± 447.0	532.3 ± 1328.5
Mean ± SEM, Eosinophils in peripheral blood (%)	2.6 ± 1.3	3.9 ± 2.5	2.9 ± 1.4	6.8 ± 4.7 **^,#,†††^
Mean ± SEM, Total number of Eosinophils in peripheral blood	138.2 ± 57.1	228.9 ± 163.5	156.8 ± 102.2	420.1 ± 331.6 **^,#,††††^

**; *p* < 0.01, vs. control uncinate tissue (UT), ^#^; *p* < 0.05, vs. chronic rhinosinusitis (CRS) (UT), ^†††^; *p* < 0.001, vs. Non-ECRS (nasal polyps (NPs)), ^††††^; *p* < 0.0001, vs. Non-ECRS (NPs).

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
