# Peer review of "Enhanced 15-Lipoxygenase 1 Production is Related to Periostin Expression and Eosinophil Recruitment in Eosinophilic Chronic Rhinosinusitis"

_biomolecules, 2020, doi:10.3390/biom10111568_

Round 1

Reviewer 1 Report

Yoshimasa Imoto et al are reporting in a manuscript the importance of the 15-lipoxygenase production on the eosinophilic infiltration and subsequent tissue inflammation into nasal polyps. To do that, the authors quantified directly in NP tissues of several subgroups of patients and controls the levels of mRNA coding for 15-LOX-1 and periostin, localized the expression of the enzyme by immunofluorescence directly in tissue eosinophils and finally used an eosinophilic cell line to explore the mechanism of production of 15-LOX-1. The manuscript is focusing on 15-LOX-1 and eosinophils in nasal polyps. It is very well written, the results are clearly described and presented in the figures. 

  Major comments: > Figure 1: ECRS group has 48 subjects, and the levels measured for 15-LOX-1 mRNA are ranging from 5 to 60. This high range makes wonder if the blood eosinophil count can be related with the levels of 15-LOX-1. The authors should add the Absolute eosinophil count for each group in the table 1 if known. Were the asthmatics subjects under treatment? If yes, would it be interesting to mention the ratio of responder vs non responder?     > Figure 4 seems to show individual values for all subjects from the 4 groups.     As periostin is related to eosinophil recruitment (cf discussion) and 15-LOX-1 is not eosinophil specific, showing in Figure 4B a strong significant correlation between the transcriptional levels of periostin and 15-LOX-1 supports the idea that eosinophils are a significant source of 15-LOX-1 in NPs. Have the authors looked at the correlations between 15-LOX-1-Eos, and Periostin-Eos in the ECRS group only? Are they also significant?
Figure 4 needs to show the patients subgroups, or colour the ECRS individual values. Authors can decide, based on the messages, to plot all subjects, or just 1 or 2 groups, but the different R and P should be written if the ECRS group alone shows significant correlation.  Are the levels of periostin also significantly correlated with the eosinophils number in NPs? if yes, it should be mentioned in the text.
  Minor comments: >Table 1 : Are the values shown for age, IgE and Eo: mean+/- SEM? What is the unit for total IgE?   >The authors need to indicate the n for each figure panel presented.
  >Figure 5 and 6, in vitro culture experiments. Unless the variables are shown to be normally distributed, please use a Wilcoxon signed rank test to compare the data. Have other mediators like IL-4 tested on Eol-1?   >In Figure 1,5 and 6: Please rename the y axis 15-LOX-1, and not 15-LO.   >The introduction highlights the presence of two phenotypes for CRSwNP, neutrophilic or eosinophilic-dominant inflammation.  It could be of interest to mention in the discussion the differences that exist between the two granulocytes regarding the use of 15-LO enzymes. Eosinophils are predominantly using 15-LOX-1 and neutrophils 15-LOX-2. I understand the paper focuses on eosinophils only, but patients from Asia are more likely to have neutrophils involved. A suggested citation could be the work of Archambault S. et al. Plos One 2018.

Author Response

Thank you very much for reviewing our manuscript and offering valuable comments. We have revised the manuscript accordingly and addressed your comments with a point-by-point response.

Comments and Suggestions for Authors

Yoshimasa Imoto et al are reporting in a manuscript the importance of the 15-lipoxygenase production on the eosinophilic infiltration and subsequent tissue inflammation into nasal polyps. To do that, the authors quantified directly in NP tissues of several subgroups of patients and controls the levels of mRNA coding for 15-LOX-1 and periostin, localized the expression of the enzyme by immunofluorescence directly in tissue eosinophils and finally used an eosinophilic cell line to explore the mechanism of production of 15-LOX-1. The manuscript is focusing on 15-LOX-1 and eosinophils in nasal polyps. It is very well written, the results are clearly described and presented in the figures.

Major comments:

Figure 1: ECRS group has 48 subjects, and the levels measured for 15-LOX-1 mRNA are ranging from 5 to 60. This high range makes wonder if the blood eosinophil count can be related with the levels of 15-LOX-1. The authors should add the Absolute eosinophil count for each group in the table 1 if known.

Response:

We appreciate the reviewer’s suggestion. We added the absolute eosinophil count in Table 1.

We are sorry that the number of ECRS subjects is 47, not 48. We also showed correct number in Table 1.

Were the asthmatics subjects under treatment? If yes, would it be interesting to mention the ratio of responder vs non responder?

Response:

All subjects were prohibited from taking oral steroids for at least 4 weeks prior to surgery. Although several patients showed comorbidity of asthma, none of subjects required oral corticosteroid. We described the details of patients (line 77 to 86).

Figure 4 seems to show individual values for all subjects from the 4 groups. As periostin is related to eosinophil recruitment (cf discussion) and 15-LOX-1 is not eosinophil specific, showing in Figure 4B a strong significant correlation between the transcriptional levels of periostin and 15-LOX-1 supports the idea that eosinophils are a significant source of 15-LOX-1 in NPs. Have the authors looked at the correlations between 15-LOX-1-Eos, and Periostin-Eos in the ECRS group only? Are they also significant?

Response:

The reviewer raises a very reasonable point. We analyzed the data again about ECRS group, and we found significant correlation between the expression of 15-LOX-1 and the number of eosinophils (r = 0.4102, p < 0.01), as well as the expression of periostin and the number of eosinophils in NPs (r = 0.8297, p < 001). We added each result in Figure 4 and the sentences (line 224 to 233).

Figure 4 needs to show the patients subgroups, or colour the ECRS individual values. Authors can decide, based on the messages, to plot all subjects, or just 1 or 2 groups, but the different R and P should be written if the ECRS group alone shows significant correlation.  Are the levels of periostin also significantly correlated with the eosinophils number in NPs? if yes, it should be mentioned in the text.

Response:

We added the new figures that show both all subjects and ECRS subjects. We also showed the different R and P value in ECRS group in Figure 4. As reviewer pointed out, we found that there was a significant correlation between the expression levels of periostin and the number of eosinophils in NPs (r = 0.588, p < 0.0001, n = 71). We mentioned these results in the text (line 233 to 234).

Minor comments:

Table 1 : Are the values shown for age, IgE and Eo: mean+/- SEM? What is the unit for total IgE?  

Response:

We appreciate the reviewer’s suggestions. All data are presented as the mean ± standard error of the mean (SEM) as shown in the Statistical analysis section. We also added the same sentence in Patient recruitment and clinical sample collection section. The unit for total IgE is IU/ml. We added the value in Table 1.

The authors need to indicate the n for each figure panel presented.

Response:

We added the number of subjects in each figure.

Figure 5 and 6, in vitro culture experiments. Unless the variables are shown to be normally distributed, please use a Wilcoxon signed rank test to compare the data. Have other mediators like IL-4 tested on Eol-1?

Response:

We appreciate the reviewer’s suggestion. We re-analyzed about the in vitro results by Wilcoxon signed rank test, and we found the significant p-value. We showed the sentences (line 160). We also stimulated EoL-1 cells by IL-5, however, we did not find any effects on enhancement of 15-LOX-1. We added the results (line 266, Figure 6B).

In Figure 1,5 and 6: Please rename the y axis 15-LOX-1, and not 15-LO.

Response:

We corrected the y-axis label as “ 15-LOX-1“.

The introduction highlights the presence of two phenotypes for CRSwNP, neutrophilic or eosinophilic-dominant inflammation.  It could be of interest to mention in the discussion the differences that exist between the two granulocytes regarding the use of 15-LO enzymes. Eosinophils are predominantly using 15-LOX-1 and neutrophils 15-LOX-2. I understand the paper focuses on eosinophils only, but patients from Asia are more likely to have neutrophils involved. A suggested citation could be the work of Archambault S. et al. Plos One 2018.

Response:

The reviewer raises a very significant point. To date, it has been considered that phenotypes of CRSwNP in Asia seems to be more neutrophilic, however, JESREC study from Japan revealed that the phenotype of CRSwNP in Japan was more eosinophilic compared to other Asian countries. As reviewer suggested, the differences of 15-LOX type between neutrophils and eosinophils may reflect our results. We are very grateful to reviewer, and we added the comments in the discussion section (line 288 to 289).

Reviewer 2 Report

This interesting paper suggests that 15-LOX-1mRNA in nasal polyps was significantly higher in eosinophilic CRS patients than in non- eosinophilic patients and correlated with periostin levels.

Immunofluorescence showed that airway epithelial cells and eosinophils in NPs expressed 15-LOX-1.

Interleukin-33 enhanced 15-LOX-1 expression in a tumour cell line, Eol-1 cells. 

The paper needs amendment however.

Abstract Conclusion -  "may be used to distinguish ECRS and non-ECRS"  is not true. There is clearly an overlap in 15LOX-1 levels and no ROC curve is shown.

Introduction- line 38 uses Th2, line 40 Type 2. it should probably be the latter.

Methods 

Patients- atopy? ( another source of Th2 cytokines).

             -medication?( corticosteroids reduce eosinophils).

             -AERD- how determined?

             - any ANCA positive subjects in the ECRS group?

The following are missing: details of peripheral blood eosinophil measurement, details of periostin measurement, source of IL-33, details of immunohistochemical analysis- who read the slides, ? blinded etc. 

Line 134 Not a "varied concentration" but presumably "varying concentrations".

Results

Did peripheral blood eosinophils correlate with ECRS status? Or with 15-LOX1 levels?

If possible the AERD subjects should be indicated in Fig 1.

Figure 2. There is staining in B not in the epithelium- where is this?

Perhaps the immunohistochemistry results could be quantified, rather than just showing representative pictures.

Cell culture- why was IL5 not used? 

Cell line- why only IL-33?

Discussion

 Evidence from the 2 papers mentioned below should be included.

The"present finding may also have significant value for  choosing a therapeutic strategy." The paper does not actually support this as the authors have  omitted the use of some molecules such as IL-5, known to be of importance in nasal polyposis.

Line 260- ST2 needs explanation

References

The following papers and their findings should be quoted:

 Nishida M et al.,Diagnostics 2020, 10, 246; doi:10.3390/diagnostics10040246

Li Z et al. Nasal polyps 15-lipoxygenase 1 promotes CCL26/Eotaxin3 expression through ERK activation.July 2019.Journal of Allergy and Clinical Immunology 144(5)DOI: 10.1016/j.jaci.2019.06.037

Author Response

Thank you very much for reviewing our manuscript and offering valuable comments. We have revised the manuscript accordingly and addressed your comments with a point-by-point response.

Comments and Suggestions for Authors

This interesting paper suggests that 15-LOX-1mRNA in nasal polyps was significantly higher in eosinophilic CRS patients than in non- eosinophilic patients and correlated with periostin levels. Immunofluorescence showed that airway epithelial cells and eosinophils in NPs expressed 15-LOX-1. Interleukin-33 enhanced 15-LOX-1 expression in a tumour cell line, Eol-1 cells. The paper needs amendment however.

Abstract Conclusion

-  "may be used to distinguish ECRS and non-ECRS"  is not true. There is clearly an overlap in 15LOX-1 levels and no ROC curve is shown.

Response:

We appreciate the reviewer’s suggestion. We deleted the sentence “may be used to distinguish ECRS and non-ECRS”.

Introduction- line 38 uses Th2, line 40 Type 2. it should probably be the latter.

Response:

We appreciate the reviewer’s suggestion. We corrected “type 2”, not Th2.

Methods

Patients- atopy? ( another source of Th2 cytokines).

        -medication?( corticosteroids reduce eosinophils).

        -AERD- how determined?

        - any ANCA positive subjects in the ECRS group?

Response:

We described the details of patients in the Materials and Methods section (line 77 to 86). We determined AERD patients described as followings and added the sentences (line 83 to 86);

AERD was diagnosed based upon clinical criteria and was defined by the presence of asthma and at least one hypersensitivity reaction, including nasal congestion, or shortness of breath within 2-3 hour of ingestion of either aspirin or NSAIDs.

The following are missing: details of peripheral blood eosinophil measurement, details of periostin measurement, source of IL-33, details of immunohistochemical analysis- who read the slides, ? blinded etc.

Response:

We appreciate for the reviewer’s comments. Whole blood cells were collected with EDTA before surgery to count the proportions of white blood cells (line 86 to 87). The mRNA expression of periostin was measured by real time PCR described in line 108. As reviewer suggested, we analyzed the immunohistochemistry results by semi-quantified analysis shown in Figure 2E. For the semiquantitative analysis of 15-LOX-1, slides were blinded, 10 photographic fields were randomly taken from each slide and then rated on a scale of 0 to 3 by a blinded observer. A rating of 0 indicated no staining, 1 indicated mild staining, 2 indicated moderate staining, and 3 indicated intense staining. We added the results and method (line 123 to 126, and line 193 to 197, Figure 2E).

Line 134 Not a "varied concentration" but presumably "varying concentrations".

Response:

We corrected “varying concentration” not varied concentration.

Results

Did peripheral blood eosinophils correlate with ECRS status? Or with 15-LOX1 levels?

Response:

We looked again about the correlations between peripheral blood eosinophils and 15-LOX-1 expression levels, and found that there was a significant correlation (r = 0.4428, p < 0.001). We added the results in the results section (line 225 to 227).

If possible the AERD subjects should be indicated in Fig 1.

Response:

As reviewer pointed out, showing results of AERD subjects would emphasize significant value for the manuscript. We corrected Figure 1 to show the results of AERD subjects.

Figure 2. There is staining in B not in the epithelium- where is this?

Response:

Because much eosinophils infiltrate into sub-epithelial layer in NPs of ECRS, we observed sub-epithelial lesion to detect eosinophils. We added the “sub-epithelial lesion of NP tissues” in Figure 3 legend (line 215).

Perhaps the immunohistochemistry results could be quantified, rather than just showing representative pictures.

Response:

As reviewer suggested, we analyzed the immunohistochemistry results by semi-quantified analysis shown in Figure 2E. Details of the method was also added in the Immunohistochemistry section (line 123 to 126, and line 193 to 197).

Cell culture- why was IL5 not used?

Response:

We also stimulated EoL-1 cells by IL-5, however, we did not find any effects on enhancement of 15-LOX-1. We added the results (line 266, Figure 6B).

Cell line- why only IL-33?

Response:

Although there are many cytokines and chemokines contribute to the pathogenesis of CRSwNP, we thought that interaction between epithelial cells and eosinophils may be one the most important mechanisms. IL-33 can be released form epithelial cells by several trigger include fungi, bacteria, and antigens. As there had not been mentioned about the effects of IL-33 towards the 15-LOX-1 so far, we wanted to investigate whether IL-33 can enhance 15-LOX-1 in eosinophils. We also stimulated EoL-1 cells by IL-5, however, we did not find any effects on enhancement of 15-LOX-1 (line 266, Figure 6B).

Discussion

Evidence from the 2 papers mentioned below should be included.

Response:

We have already added the Li’s manuscript. Although the reviewer suggested to include Nishida’s manuscript, their findings are mainly focused on LOX-1 and hypoxia. They demonstrated the LOX-1 levels by comparing CRSwNP and CRSsNP subjects. They did not discussed about non-ECRS and ECRS subjects. Their results are very interesting and reasonable in the pathogenesis of CRS, however, as containing their results may confuse the discussion, we refrained from quoting their manuscript.

The"present finding may also have significant value for choosing a therapeutic strategy." The paper does not actually support this as the authors have omitted the use of some molecules such as IL-5, known to be of importance in nasal polyposis.

Response:

We appreciate the reviewer’s suggestion. As reviewer pointed out, we did not observe every types of cytokine that could be potential strategy for CRS. In our results, we focused and found the effects of IL-4 and IL-33 to upregulate 15-LOX-1. Thus, we deleted the sentences “The present finding may also have significant value for choosing a therapeutic strategy”.

Line 260- ST2 needs explanation

Response:

We are grateful for the comments. We added the sentences to explain ST2 as following;

IL-33 can activate many types of immune cells via its receptor, interleukin receptor-like 1 (IL1RL1, also known as ST2) (line 308 to 309).

References

The following papers and their findings should be quoted:

Nishida M et al.,Diagnostics 2020, 10, 246; doi:10.3390/diagnostics10040246

Li Z et al. Nasal polyps 15-lipoxygenase 1 promotes CCL26/Eotaxin3 expression through ERK activation.July 2019.Journal of Allergy and Clinical Immunology 144(5)DOI: 10.1016/j.jaci.2019.06.037

We have already added the Li’s manuscript. Although the reviewer suggested to include Nishida’s manuscript, their findings are mainly focused on LOX-1 and hypoxia. They demonstrated the LOX-1 levels by comparing CRSwNP and CRSsNP subjects. They did not discussed about non-ECRS and ECRS subjects. Their results are very interesting and reasonable in the pathogenesis of CRS, however, as containing their results may confuse the discussion, we refrained from quoting their manuscript.

Round 2

Reviewer 2 Report

This paper has been improved to my satisfaction.